# Identifying Genetic Biomarkers Predicting Response to Anti-Vascular Endothelial Growth Factor Injections in Diabetic Macular Edema

**DOI:** 10.3390/ijms23074042

**Published:** 2022-04-06

**Authors:** Rajya L. Gurung, Liesel M. FitzGerald, Ebony Liu, Bennet J. McComish, Georgia Kaidonis, Bronwyn Ridge, Alex W. Hewitt, Brendan J. Vote, Nitin Verma, Jamie E. Craig, Kathryn P. Burdon

**Affiliations:** 1Menzies Institute for Medical Research, University of Tasmania, Hobart, TAS 7000, Australia; liesel.fitzgerald@utas.edu.au (L.M.F.); bennet.mccomish@utas.edu.au (B.J.M.); alex.hewitt@utas.edu.au (A.W.H.); 2Department of Ophthalmology, Flinders Health and Medical Research Institute, Flinders University, Adelaide, SA 5042, Australia; ebony.liu2@flinders.edu.au (E.L.); georgia.kaidonis@gmail.com (G.K.); bronwyn.ridge@sa.gov.au (B.R.); jamie.craig@flinders.edu.au (J.E.C.); 3School of Medicine, University of Tasmania, Hobart, TAS 7000, Australia; eye.vote@me.com (B.J.V.); nitver3@gmail.com (N.V.)

**Keywords:** anti-vascular endothelial growth factor, diabetic macular edema, genome-wide association

## Abstract

Intraocular anti-vascular endothelial growth factor (VEGF) therapies are the front-line treatment for diabetic macular edema (DME); however, treatment response varies widely. This study aimed to identify genetic determinants associated with anti-VEGF treatment response in DME. We performed a genome-wide association study on 220 Australian patients with DME treated with anti-VEGF therapy, genotyped on the Illumina Global Screening Array, and imputed to the Haplotype Reference Consortium panel. The primary outcome measures were changes in central macular thickness (CMT in microns) and best-corrected visual acuity (BCVA in ETDRS letters) after 12 months. Association between single nucleotide polymorphism (SNP) genotypes and DME outcomes were evaluated by linear regression, adjusting for the first three principal components, age, baseline CMT/BCVA, duration of diabetic retinopathy, and HbA1c. Two loci reached genome-wide significance (*p* < 5 × 10^−8^) for association with increased CMT: a single SNP on chromosome 6 near *CASC15* (rs78466540, *p* = 1.16 × 10^−9^) and a locus on chromosome 12 near *RP11-116D17.1* (top SNP rs11614480, *p* = 2.69 × 10^−8^). Four loci were significantly associated with reduction in BCVA: two loci on chromosome 11, downstream of *NTM* (top SNP rs148980760, *p* = 5.30 × 10^−9^) and intronic in *RP11-744N12.3* (top SNP rs57801753, *p* = 1.71 × 10^−8^); one near *PGAM1P1* on chromosome 5 (rs187876551, *p* = 1.52 × 10^−8^); and one near *TBC1D32* on chromosome 6 (rs118074968, *p* = 4.94 × 10^−8^). In silico investigations of each locus identified multiple expression quantitative trait loci and potentially relevant candidate genes warranting further analysis. Thus, we identified multiple genetic loci predicting treatment outcomes for anti-VEGF therapies in DME. This work may potentially lead to managing DME using personalized treatment approaches.

## 1. Introduction

Diabetic macular edema (DME) is the most common cause of visual impairment in patients with diabetes and can affect patients with either Type 1 (T1) or Type 2 (T2) diabetes mellitus (DM). It is the leading cause of central vision loss in the working-age population with a prevalence of 1.4–12.8% in patients with diabetes [1]. Although the exact pathogenesis of DME remains elusive, there is strong evidence that it develops due to complex interactions between environmental and genetic factors [2,3]. It is characterized by the accumulation of fluid, lipids, and/or hemorrhages in the macula (the part of the retina responsible for central vision) due to a breakdown of the blood–retinal barrier [4]. Angiogenic factors, such as vascular endothelial growth factor (VEGF), and inflammatory mediators have been postulated to be involved in the development of DME [5]. Since the advent of the first intravitreal anti-VEGF therapy in 2005 [6], the development of newer agents targeting the VEGF/VEGF-receptor has improved treatment outcomes for DME patients [7,8]. However, one-third of patients have an insufficient response to these currently available therapies [9,10]. A possible and significant consequence of an insufficient response is that the patient may suffer from ongoing macular injury at a molecular level, thereby leading to irreversible vision loss [11]. Several randomized control trials (RCTs) and clinic-based studies have attempted to identify ocular and systemic predictors of response to anti-VEGF treatment but there has been little research into the potential role of genetics [12,13]. There are only a few studies that have specifically aimed to identify genetic variants that distinguish between DME patients that do respond (“responders”) and do not respond (“non-responders”) to anti-VEGF injections [14,15,16,17,18,19]. Most, if not all, of these studies are candidate gene studies, with a small sample size, short follow-up period, and poorly defined response criteria. Therefore, we undertook a genome-wide association pilot study (GWAS) to explore genetic determinants of anti-VEGF treatment response in patients with DME.

## 2. Results

### 2.1. Cohort Characteristics

A total of 248 diabetic (T1 + T2) patients receiving anti-VEGF injections were identified with complete clinical data. Of these, genome-wide genotyping data were available for 234 individuals. After quality control (QC), 220 individuals with data for 2,581,674 autosomal SNPs were available for analysis. The details of the individual level and single nucleotide polymorphism (SNP) level QC are provided in the Materials and Methods section and Appendix A. For these 220 patients, the mean change in central macular thickness (CMT) was −55.17 microns and the mean change in best-corrected visual acuity (BCVA) was 3.41 Early Treatment Diabetic Retinopathy Study letters (ETDRS letters) following 12 months of anti-VEGF treatment, indicating significant improvements in vision and macular thickness following treatment (*p* < 0.05 for both measures). Demographic information and clinical parameters of the study cohort are described in Table 1.

### 2.2. Association Analysis

#### 2.2.1. Genome-Wide Significant Loci

Six SNPs at two loci showed genome-wide significant evidence for association with change in CMT, *p* < 5 *×* 10^−8^ (Figure 1A; Table 2). These included a single SNP, rs78466540, near cancer susceptibility candidate 15 (*CASC15*) gene on chromosome 6 (Appendix A) and five SNPs, rs11614480 (lead SNP), rs11615848, rs11614887, rs11615870, rs11615833, in linkage disequilibrium (LD) near *RP11-116D17.1* on chromosome 12 (Appendix A). The details of all SNPs at each significant or suggestive locus are given in Appendix A: SNP.xlsx (Sheet-1) and the corresponding Quantile–Quantile (Q–Q) plot in Appendix A.

Four loci reached genome-wide significance for association with change in BCVA (Figure 1B, Table 2). The most significant locus was represented by two SNPs on chromosome 11, rs148980760 (lead SNP) and rs117744949 near the Neurotrimin (*NTM*) gene. A second locus on chromosome 11, rs57801753 in the *RP11-744N12.3* gene, a single SNP on chromosome 5, rs187876551 near Phosphoglycerate Mutase 1 Pseudogene 1 (*PGAM1P1*), and a locus on chromosome 6, rs118074968 near theTBC1 Domain Family Member 32 (*TBC1D32*) also reached significance. Details of all SNPs are given in Appendix A: SNP.xlsx (Sheet-2) and the corresponding Q–Q plot in Appendix A and the regional plots as shown in Appendix A.

#### 2.2.2. Suggestive Loci

We also identified several other loci with *p* values that were suggestive of association, but that did not reach genome-wide significance (≤5 × 10^−7^). Appendix A**:** SNP.xlsx (sheet 1–2) summarizes the lead SNPs found in each of these suggestive loci with details of SNPs in LD. There were two suggestive loci for change in CMT: chromosome 4, EPH Receptor A5 (*EPHA5*) gene, lead SNP rs139041797, and chromosome 16, Deoxyribonuclease I (*DNASE1*) gene, lead SNP rs35275535. For change in BCVA, there were five suggestive loci: chromosome 11, Apolipoprotein O 3733.1 (*APOO3733.1*) gene, lead SNP rs78772036; chromosome 11, *RP11-179A16.1* gene, lead SNP rs75537672 and four other individuals SNPs (rs17786210, rs76805698, rs77694097, rs117434848).

#### 2.2.3. Conditional Analysis

Conditional analyses were applied to identify secondary association signals across the entire genome. For the change in CMT, analysis was performed conditioned on the lead SNPs, rs78466540, and rs11614480. For the change in BCVA, analysis was performed conditioned on the lead SNPs rs148980760, rs187876551, rs57801753, and rs118074968. No additional SNPs reached genome-wide or suggestive significance thresholds for either of the outcomes. The resulting Manhattan and Q–Q plots for the conditional analyses are presented in Appendix A.

#### 2.2.4. Logistic Regression Model

A logistic regression model (responders vs. non-responders) was implemented stratifying outcomes on the basis of the functional and anatomical response. The baseline and clinical characteristics of responders and non-responders for both outcomes are shown in Appendix A. Adjustments for relevant covariates were done based on the baseline clinical characteristics for each outcome. No variants reached genome-wide or suggestive significance thresholds for either the functional or anatomical outcome (Appendix A).

## 3. Functional Annotation

### Genome-Wide Significant Loci

The results of functional annotation for the genome-wide significant loci are given in Table 3. For the change in CMT, the lead SNP, rs11614480 had a Combined Annotation-Dependent Depletion (CADD) score of 18. Expression quantitative trait locus (eQTL) analysis showed that this SNP was associated with *RP11-116D17.3* gene expression in several tissues in the Genotype-Tissue Expression (GTEx) database (Table 3). Furthermore, eQTL analysis showed a significant association between the *RP11-116D17.3* gene and five nearby SNPs in LD with the lead SNP (rs721107, rs11610611, rs11610643, rs11611628, rs11609330) corresponding to multiple tissues including the pancreas and thyroid [Appendix A: SNP.xlsx (Sheet 1 and 3)]. Of these, rs11610611 had a CADD score of 14.36 and rs11611628 a RegulomeDB score of 2a. For the change in BCVA, only rs118074968 and nearby SNPs in LD exhibited an eQTL effect for the Gap Junction Protein Alpha 1 (*GJA1*) gene in a single tissue [Table 3 and Appendix A: SNP.xlsx (Sheet 2 and 3)]. 

We also performed eQTL analysis for significant loci in retinal tissue using the Eye Genotype Expression (EyeGEx) database; however, no associations were observed between these variants and the expression of retinal genes [Table 3 and Appendix A**:** SNP.xlsx (Sheet 1–3)].

## 4. Ocular Tissue Database

We queried the candidate genes for expression in different eye tissues and cells in the Ocular Tissue Database (OTDB), with results given in Appendix A. A total of six genes were expressed in the OTDB. Of the six genes, the Probe Logarithmic Intensity Error score (PLIER score) was highest for the *GJA1* gene (PLIER = 887.985) corresponding to choroid/retinal pigment epithelium. Regarding retinal tissue expression, the score was highest for the *NTM* gene (PLIER = 73.7647), followed by the *GJA1* gene (PLIER = 61.0931).

## 5. Discussion

We identified multiple genome-wide significant loci associated with anti-VEGF response in DME, defined as either change in CMT or BCVA. After anti-VEGF injection, individuals carrying the alternate allele of rs78466540 (G) or rs11614480 (C) are more likely to be non-responders measured by the change in CMT. Likewise, those carrying the alternate allele of rs148980760 (C), rs187876551 (A), rs57801753 (C), or rs118074968 (G) are more likely to be non-responders measured by the change in EDTRS letters post-anti-VEGF injection. Two of the lead SNPs, rs11614480 and rs118074968, exhibited eQTL effects for the *RP11-116D17.3* and *GJA1* genes, respectively.

The locus corresponding to the *CASC15* gene (chromosome 6; rs78466540) is associated with a sub-optimal reduction in CMT. *CASC15* is a long intergenic non-coding RNA (lincRNA). This gene is involved in cellular programming, including proliferation, migration, and angiogenesis, in cancer cells via the hypoxia-induced VEGF pathway [20]. VEGF is a key player in the pathogenesis of diabetic retinal complications including DME [5] and is the molecular target of the anti-VEGF therapies under investigation here. However, as shown in the regional plot Appendix A, this locus has very few supporting SNPs in LD and could be a false positive association. Consequently, the locus requires replication in additional datasets, as do all the findings reported here.

Another locus of interest is on chromosome 12 (lead SNP: rs11614480). This locus harbored several genome-wide significant SNPs, all in LD with the lead SNP, and is further supported by many SNPs showing suggestive associations with change in CMT. There is no definitive information available for the gene closest to this locus, *RP11-116D17.1*. This is a common finding for many GWAS [21,22], which often implicate gene deserts, genes of unknown function or with no HGNC symbol [23], or of previously unsuspected disease relevance. Most require replication in an independent population, and/or validation using other strategies, such as fine mapping and lab-based functional studies. It is also well known that the nearest gene to an association signal is not necessarily the causal gene [24]. Interestingly, this locus is a significant eQTL for *RP11-116D17.3*, which is also near to the SNPs and is expressed in many biologically plausible tissues relevant to diabetes, including pancreas, adipose tissue, thyroid, pituitary, and adrenal gland. This locus is thus a plausible candidate for future studies. 

Another noteworthy locus is on chromosome 11 (lead SNP: rs148980760). The closest gene to the lead SNP is *NTM*, a member of the IgLON family that has been previously linked to late-onset Alzheimer’s disease, a neurodegenerative disorder [25]. Interestingly, akin to disorders such as Alzheimer’s, the pathogenesis of DME is now considered part of a spectrum of a more comprehensive neurovascular and neurodegenerative process rather than a strictly vascular phenomenon [26]. 

Our eQTL analysis of significant loci also revealed some interesting genes: *GJA1* and *RP11-116D17.3*. The *GJA1* gene, corresponding to eQTL rs118074968 (chr6), has been previously implicated in oculodentodigital syndrome [27,28]. The *RP11-116D17.3* gene has not been previously associated with disease, but as noted, this gene is expressed in several disease-relevant tissues. However, none of our significant loci were eQTLs for more retina-specific tissues in the EyeGEx database. Nonetheless, the *GJA1* gene is highly expressed in the choroid, retinal pigment epithelium, and retinal tissue as observed in the OTDB (Appendix A). The EyeGEx database does not differentiate distinct retinal layers or tissues, thus a lack of an association does not necessarily mean that there are no eQTLs. Although the EyeGEx database contains data from a substantial number of donors, a more comprehensive eQTL database including distinct layers of the retina would add significant value to the interpretation of our findings. 

Response to anti-VEGF injections is measured by changes in BCVA and/or CMT. Given the use of both measurements in clinical decision making, both were used as outcomes in this study. While we also categorized our response measures into a case-control design using several vision and macular thickness criteria, we did not find any significant or suggestive loci for these phenotypes. Previous studies have shown that stratifying continuous data into categories can lead to information bias and loss of statistical power [29,30], the latter of which is likely to have affected our study, particularly due to its small sample size. 

The response to anti-VEGF injection is likely affected by many clinical, genetic, and environmental factors. Our analyses have adjusted for several relevant confounding clinical factors including age, HbA1c, duration of DR, injection type, and baseline BCVA or CMT. In clinical practice, poor compliance leading to inadequate injection frequency is another important cause of the poor response. The frequency of injections in our study (8 ± 3) was similar to previous RCTs for DME [31,32] and slightly higher than in other real-world studies [33,34]; therefore, it is unlikely that undertreatment was a confounding factor. Furthermore, as the genomic control inflation factor λ was 1.000, our analyses suitably controlled for population stratification and cryptic relatedness. We included both T1 and T2 diabetes patients under the hypothesis that response to treatment for ocular complications would be driven by different factors than those that cause the sub-types of diabetes; stratified analyses would require a larger cohort. The two main limitations of this study were the small sample size and the lack of a replication cohort. However, we believe this is the first GWAS to date looking at the response to anti-VEGF treatment for DME and is one of only a handful of studies that have investigated the genetic determinants or molecular responses to anti-VEGF injections in DME [14,15,16,17]. These prior studies also had sample size limitations, were focused on candidate genes, or were evaluating gene expression responses to treatment. Although the sample size is a limitation, this study has the largest DME anti-VEGF treatment cohort for a genetic study to date. Importantly, strong genetic effects can be detected with small but phenotypically homogenous samples, as demonstrated in our homogenous group of OCT-diagnosed DME patients with visual impairment. Next, owing to a retrospective design, a more detailed data collection regarding OCT biomarkers was not possible. Finally, no single type of anti-VEGF drug was used consistently in this study. Many patients received two or three different anti-VEGF agents over the course of 12 months, reflecting real-world practice. It is likely that the patients were switched to a different anti-VEGF injection because they were not responding to their current agent, as has been reported in similar studies [35,36]. 

## 6. Conclusions

In conclusion, this study has identified several putative genetic variants that predict anti-VEGF treatment response. Given the relatively small cohort size, these associations should be considered provisional and replicated in adequately powered independent studies. International multi-center collaborations, including different ethnicities, will further advance our understanding of the biological pathways underpinning the treatment response of DME patients. Such studies may enable the development of novel therapies and the prediction of who will benefit from specific treatment modalities as we move towards precision medicine for the treatment of DME.

## 7. Materials and Methods

### 7.1. Study Design and Participants

This was a retrospective multi-center study involving multiple eye clinics in the Australian states of South Australia and Tasmania. All participants were enrolled through the Tasmanian Ophthalmic Biobank Study (University of Tasmania) or the Genetic Risk Factors in Complications of Diabetes Study (Flinders University). The eligibility criteria included T1 and T2 DM patients ≥18 years who had received any intravitreal anti-VEGF injections (Aflibercept, Regeneron; Bevacizumab, Genentech; Ranibizumab, Novartis) between 2013 and 2020 for the treatment of DME. DME cases were defined as those with clinically diagnosed center-involving DME and confirmed by central macular thickness (CMT ≥ 315 microns) measured by optical coherence tomography (OCT). Eyes with cysts in the central 1000 microns or any intraretinal or subretinal fluid were included in this study, independent of the CMT parameter. Exclusion criteria were: (a) any vitreoretinal surgery, systemic anti-VEGF therapy, or intra-ocular steroid in the six months preceding the initiation of anti-VEGF injection, (b) insufficient visibility of the fundus for retinal diagnosis, (c) incomplete follow-up data, and (d) inability to give consent. The better responding eye was included as the study eye for patients receiving bilateral anti-VEGF injections. 

### 7.2. Phenotyping

Twelve months of retrospective data were collected from each participant’s medical record. The data included clinical and demographic characteristics (Table 1). Treatment decisions, including choice of anti-VEGF drug, switching between agents, re-treatment criteria, and treatment interval, were at the discretion of the treating physician and varied between clinics and doctors. BCVA, CMT, drug used, and injection number were recorded at baseline and at each injection visit. For statistical analysis, Snellen BCVA was converted to ETDRS letter scores [37].

### 7.3. Outcome Measures

The primary outcome measures were the change in CMT and BCVA at 12 months after treatment onset. The change was defined as the baseline CMT or BCVA before treatment subtracted from the final CMT or BCVA after 12 months and was analyzed as a quantitative variable. The data were further categorized into functional and anatomical responder or non-responder groups for exploration of secondary outcomes. A functional responder was defined as (1) ≥5 ETDRS letters improvement from baseline [32,38] or (2) ≥15 ETDRS letters improvement from baseline [39,40]. An anatomical responder was defined as ≥10% reduction in CMT from the baseline [38,41]. We also assessed a separate model wherein a functional non-responder was defined as ≥5 ETDRS letters loss from baseline [38]. Each defined group was compared to the remainder of the cohort.

### 7.4. Genotyping and Quality Control

Genomic DNA was extracted from peripheral venous blood (Illustra-DNA-extraction kit, BACC3) and quantitated using the Qubit Fluorometer (Thermo Fisher Scientific). A total of 234 DNA samples were genotyped on one of three different versions of the Illumina Global Screening Array SNP genotyping platform: GSAMD-24v3-0-EA_A1, GSA-24v1-0_C1, or GSA-24v3-0-A1. Genotypes were called using Genome Studio (version 2.0, Illumina, United States). QC of genotype data was performed using the Genome Studio genotyping module as described in Gua et al. [42]. Briefly, for each version of the array, poor quality markers (genotyping failed in >1% of samples) and poor quality DNA samples (genotyping failed for >1% of the markers) were removed. Additional QC procedures were undertaken as described in the Illumina genotyping technote, https://www.illumina.com/Documents/products/technotes/technote_infinium_genotyping_data_analysis.pdf (accessed on 18 October 2020); *filtering of haploid markers:* sex chromosome, mitochondrial, pseudo-autosomal; *filtering of autosomal markers:* cluster separation, AB R Mean, AB T Mean, Het Excess, Minor_Freq. The data from each array platform were then imported separately into PLINK (version 1.9/2.0) [43] where further QC was performed on each dataset separately and then again once the three datasets were merged (Appendix A). Briefly, individuals with discordant sex information, a missing genotype rate >0.02, or heterozygosity ±6 SDs from the mean were excluded. Genetically related individuals were detected by calculating pairwise identity-by-descent/king-cut off, and the individual with the lower genotyping rate in any pair (identity-by-descent greater ≥0.2/king-cut off ≥0.0884) was removed. SNPs were excluded if they had a missing genotype rate >0.02 and/or had a minor allele frequency (MAF) <0.01.

Ancestral outliers were then identified using principal component analysis (PCA) based on pruned sets of autosomal SNPs from the merged datasets, selected by an LD r^2^ threshold of 0.2 in windows of 50 SNPs and a moving step of five SNPs. By anchoring against 1000 Genomes (1KG), samples of European (EUR), Asian (ASN), Admixed American (AMR), and African (AFR) descent were identified. For our analyses, only individuals of European descent were included and individuals from other ethnicities and extreme outliers were excluded (Appendix A). Following QC, the merged GWAS dataset consisted of 221 individuals with genotyping data for 281,952 SNPs that were common to all three versions of the array.

### 7.5. Imputation

Imputation was performed using the Michigan imputation server [Minimac4 (1.5.7); https://imputationserver.sph.umich.edu/index.html#!run/minimac4] (accessed on 12 January 2021). Using the Haplotype Reference Consortium panel (HRC panel, Version r1.1 2016), the allele dosage for over two million SNPs (after QC) was imputed for the 22 autosomal chromosomes. Post imputation, SNPs were excluded according to the following parameters: imputation quality score R2 < 0.8, MAF < 0.02, missing genotype rate > 0.03, heterozygosity ±6 SDs from the mean, and HWE deviation *p* < 1 × 10^−06^ (Appendix A). This resulted in a final GWAS dataset of 220 individuals with genotyped and imputed data for 2,581,674 autosomal SNPs.

### 7.6. GWAS Statistical Analysis

Associations between SNP genotypes (including imputed genotypes) and DME outcomes (change in CMT/BCVA) were evaluated by linear regression for continuous traits and logistic regression for categorical (responder/non-responder) groupings using PLINK2.0. Evidence for association was assessed using an additive model for risk by allele dose. The resulting residuals for change in CMT/BCVA distributions were approximately normally distributed. The original scales were used when reporting effect sizes (microns/allele for change in CMT and ETDRS letters/allele for change in BCVA). Age, HbA1c, duration of diabetic retinopathy (DR), injection type, and baseline CMT/BCVA were included in the model as covariates as they are consistently described in previous clinical studies as influencing change in CMT or BCVA following anti-VEGF treatment [10,12]. The first three principal components (PCs) were also included as covariates to adjust for population stratification (Appendix A). Genomic inflation factors (λ) were calculated to evaluate any residual population stratification in PLINK2.0. Q–Q plots and Manhattan plots were created in R version 4.0.2 (http://www.R-project.org/) with the GWASTools (Bioconductor: http://www.bioconductor.org) [44] and qqman package [45], respectively. Evidence for association was set at *p* ≤5 × 10^−8^ for genome-wide significance and at *p* ≤ 5 × 10^−7^ for suggestive significance. Local/regional association plots were constructed with FUMA v1.3.6 online tool (https://fuma.ctglab.nl/) (accessed on 18 January 2021) [46]. A locus was defined as 1Mb surrounding the genome-wide significant SNP, including any SNPs in LD (r^2^ ≥ 0.4) and the gene nearest to the lead SNP. We also performed conditional analysis to determine independent association signals across the entire genome, conditioning on the genome-wide significant SNPs.

Additionally, associations between SNP genotypes and secondary outcomes (functional or anatomical response) were evaluated using logistic regression, adjusting for covariates (age, duration of DR, HbA1c, baseline CMT/BCVA, and the first three PCs). 

### 7.7. Demographic Data Statistical Analysis

For the demographic data, the mean change in BCVA and CMT from baseline to 12 months visits were conducted using paired Student t-tests in R version 4.0.2. *p* < 0.05 was considered statistically significant. Results are presented as the mean ± SD for continuous variables and as proportions (%) for categorical variables.

### 7.8. Functional Annotation

To discern key genes and variants at the identified GWAS loci, we conducted functional follow-up analyses through the FUMAgwas online portal. The following criteria incorporated in the annotation tool were applied: (1) CADD score (range 1–99) to determine the deleteriousness of the variant [47], (2) RegulomeDB score (range 1–7), for potential regulatory functions [48], and (3) eQTL analysis was performed using expression summary statistics from the GTEx database v8: http://www.gtexportal.org/home/datasets) (accessed on 18 January 2021) with a false discovery rate (FDR) < 0.05. As the GTEx database does not include ocular tissues, expression summary statistics for the candidate genes in the retina from the EyeGEx database (https://gtexportal.org/home/datasets > External datasets) (accessed on 18 January 2021) were used to evaluate whether any of our variants showed significant effects on expression levels at FDR < 0.05 [49]. Finally, we also queried our lead candidate genes from our association and eQTL analyses for expression in different ocular tissues, including the retina, using the ocular tissue database (https://genome.uiowa.edu/otdb/) (accessed on 20 January 2021) [50].

## Figures and Tables

**Figure 1 ijms-23-04042-f001:**
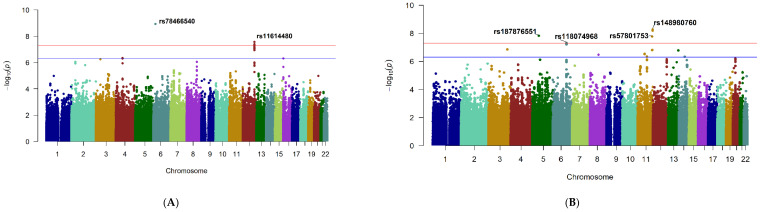
Association analyses for treatment response to anti-vascular endothelial growth factor (VEGF) therapy in diabetic macular edema (DME). The *x*-axis represents chromosomal position of each SNP, and the *y*-axis shows the -log10 (*p*-value) for association with (**A**): Change in central macular thickness (CMT) over 12 months of treatment and (**B**): Change in best-corrected visual acuity (BCVA) over 12 months. Red and blue horizontal lines correspond to the thresholds for genome-wide significant (*p* ≤ 5 × 10^−^^8^) and suggestive association (*p* ≤ 5 × 10^−^^7^), respectively.

**Table 1 ijms-23-04042-t001:** Overall baseline and clinical characteristics.

Variable	N = 220
*Patient related*	Parameter
Age (years)	66.35 (12.16)
Gender:Male, N (%)	151 (68.6)
BMI (kg/m^2^)	33.76 (8.00)
DM: T2, N (%)	181 (82.3)
DM duration (years)	22.74 (10.15)
HbA1c (mg/dl)	8.38 (1.63)
HTN: Yes, N (%)	188 (85.5)
Nephropathy: Yes, N (%)	120 (54.5)
Hyperlipidemia: Yes, N (%)	198 (90)
Smoker: Yes, N (%)	111 (50.5)
*Eye related*	
Baseline BCVA (ETDRS letters)	64.25 (14.52)
Final BCVA (ETDRS letters)	67.66 (15.54)
Change in BCVA (ETDRS letters)	3.41 (12.05) *
Baseline CMT (microns)	380.57 (104.08)
Final CMT (microns)	325.40 (75.38)
Change in CMT (microns)	−55.17 (99.16) *
Laterality: RE, N (%)	113 (51.4)
Lens status: Phakic, N (%)	141 (64.1)
DR duration (years)	7.95 (4.30)
*DR severity N (%)*	
Mild DR	45 (20.5)
Moderate DR	67 (30.5)
Severe DR	35 (15.9)
PDR	72 (32.7)
Injection number	8.02 (3.07)
*Injection type: N (%)*	
Bevacizumab	126 (57.3)
Aflibercept	25 (11.4)
Ranibizumab	37 (16.8)
Mixed	32 (14.5)

Abbreviations: BCVA = best corrected visual acuity; BMI = body mass index; CMT = central macular thickness; DM = diabetes mellitus; DR = diabetic retinopathy; ETDRS = early treatment diabetic retinopathy study; HTN = hypertension; PDR = proliferative diabetic retinopathy; PRP = pan-retinal photocoagulation; RE = right eye. Data are presented as means (SD) for continuous variables and number, percentage (%) for categorical variables. * *p* < 0.05 for comparison of pre and post-treatment measures.

**Table 2 ijms-23-04042-t002:** Genome-wide significant associations.

Locus *	Chr	Position ^†^	Lead SNP	Ref	Alt	AAF	Beta ^‡^	*p* Value
Change in CMT (microns)								
*CASC15*	6	21755718	rs78466540	A	G	0.03	115.80	1.16 × 10^−09^
*RP11-116D17.1*	12	115772072	rs11614480	T	C	0.07	71.08	2.69 × 10^−08^
	12	115772088	rs11615848	G	T	0.07	71.08	2.69 × 10^−08^
	12	115772313	rs11614887	T	C	0.07	71.02	2.73 × 10^−08^
	12	115772214	rs11615870	G	T	0.07	71.02	2.73 × 10^−08^
	12	115772032	rs11615833	G	A	0.07	70.84	4.04 × 10^−08^
Change in BCVA (ETDRS letters)								
*NTM*	11	132228056	rs148980760	A	C	0.03	−17.98	5.30 × 10^−09^
	11	132237087	rs117744949	G	A	0.03	−17.50	6.57 × 10^−09^
*PGAM1P1*	5	57535905	rs187876551	G	A	0.02	−21.80	1.52 × 10^−08^
*RP11-744N12.3*	11	128524088	rs57801753	T	C	0.02	−20.19	1.71 × 10^−08^
*TBC1D32*	6	121280029	rs118074968	A	G	0.03	−16.40	4.94 × 10^−08^

Abbreviations: Alt = alternate allele; AAF = alternate allele frequency; BCVA = best corrected visual acuity; *CASC15* = cancer susceptibility candidate 15; Chr = chromosome; CMT = central macular thickness; ETDRS = early treatment diabetic retinopathy study; *NTM* = Neurotrimin; *PGAM1P1* = Phosphoglycerate Mutase 1 Pseudogene 1; Ref = reference allele; SNP = single nucleotide polymorphism; *TBC1D32* = TBC1 Domain Family Member 32. * Locus assigned to gene within or nearest to the association signal (lead SNP). † Genomic position in hg19. ‡ Unstandardized beta (Change in CMT = microns/allele; Change in BCVA = ETDRS letters/allele).

**Table 3 ijms-23-04042-t003:** Functional annotation of the lead SNPs at significantly associated loci.

Locus *	Chr	Position ^†^	Lead SNP	CADD Score	RegulomeDB Score (v1.1)	eQTL (GTEx v8)	Tissue	eGENE ^‡^	eQTL (EyeGEx)	eGENE ^║^
Change in CMT (microns)										
*RP11-116D17.1*	12	115772072	rs11614480	18.42	5	TRUE	Multiple tissues ^#^	*RP11-116D17.3*	FALSE	NA
*CASC15*	6	21755718	rs78466540	1.37	7	FALSE	NA	NA	FALSE	NA
Change in BCVA (ETDRS letters)										
*NTM*	11	132228056	rs148980760	2.39	4	FALSE	NA	NA	FALSE	NA
*RP11-744N12.3*	11	128524088	rs57801753	3.91	4	FALSE	NA	NA	FALSE	NA
*PGAM1P1*	5	57535905	rs187876551	0.40	4	FALSE	NA	NA	FALSE	NA
*TBC1D32*	6	121280029	rs118074968	7.59	6	TRUE	Skin-Not Sun Exposed	*GJA1*	FALSE	NA

Abbreviations: BCVA = best-corrected visual acuity; CADD = Combined Annotation-Dependent Depletion; *CASC15* = cancer susceptibility candidate 15; Chr = chromosome; CMT = central macular thickness; eQTL= expression quantitative trait locus; ETDRS = early treatment diabetic retinopathy study; *GJA1*= Gap Junction Protein Alpha 1; *NTM* = Neurotrimin; *PGAM1P1* = Phosphoglycerate Mutase 1 Pseudogene 1; SNP = single nucleotide polymorphism; *TBC1D32* = TBC1 Domain Family Member 32; # Multiple tissues: Adipose-Visceral, Adrenal gland, Spleen, Muscle-Skeletal, Heart-Left Ventricle, Atrial Appendage. * Locus assigned to gene within or nearest to the association signal (lead SNP). † Genomic positions are based on hg19. ‡ Significant gene expression corresponding to GTEx database. ║ Significant gene expression corresponding to EyeGEx database.

## Data Availability

The summary statistics for the association analyses performed in this study are available on request from the corresponding author. The raw genotype data are not publicly available due to the nature of the ethics approvals but may be shared through specific collaborative agreements. Please contact the corresponding author.

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
