# Peer review of "Identifying Genetic Biomarkers Predicting Response to Anti-Vascular Endothelial Growth Factor Injections in Diabetic Macular Edema"

_ijms, 2022, doi:10.3390/ijms23074042_

Round 1

Reviewer 1 Report

Thank you for allowing me to review this interesting manuscript.

The manuscript is well-written but hard to read, and statistical results should be clarified (below mentioned).

The authors performed a genome-wide association study to identify Association between 20 single nucleotide polymorphism (SNP) genotypes and DME outcomes (changes in central macular thickness and best-corrected visual acuity at 12 months.

Overall, the investigation presented here is good however needs to be improved text fluidity as it is a difficult subject and hard to read.

Corrections and suggestions:

  1. In the beginning please refer: to BCVA in ETDRS letters. (Line 20)
  2. “adjusting for the first three principal components, age, baseline CMT/BCVA, duration of diabetic retinopathy, and HbA1c – clarify adjustments performed. (Line 22)
  3. I suggest replacing the term collection for other more appropriate for the definition provided (line 59)
  4. The abbreviation should be written in full the first time it appears: GC (line 84)
  5. Explain why these patients were excluded based on QC. Please clarify
  6. In Table 1 – Are the following values: Baseline BCVA (ETDRS letters), Final BCVA (ETDRS letters), Change Baseline CMT (microns), and Final CMT (microns) means? Please clarify.
  7. Are the mean variation BCVA and CMT statistically significant? Please add p values to table 1.
  8. Correct to Figure 1A? (line 103).
  9. Even though this study is in the real-world environment, may you please clarify the protocol used at the hospital for the treatment of DME (including treatment regimen)?
  10. The authors define the following (line 336 to 340):

 “A functional responder was defined as

1) ≥5 ETDRS letters improvement from the baseline or

2) ≥15 337 ETDRS letters improvement from baseline BCVA.

A functional non-responder was defined as ≥5 ETDRS letters loss from baseline.

An anatomical responder was defined ≥10% reduction in CMT from the baseline.”

Please clarify, if a patient does not lose or gain vision letters would this be a responder or non-responder?

This definition should be clarified as does not follow any standard guidelines.

Also please clarify the sub responder.

  1. The authors wrote:

“Exclusion criteria were: (a) any vitreoretinal surgery, systemic anti-VEGF therapy, or intra-ocular steroid in the six months preceding the initiation of anti-VEGF injection, (b) insufficient visibility of the fundus for retinal diagnosis, (c) incomplete follow-up data, and d) inability to give consent.”

Did the authors exclude Iluvien implant (fluocinolone acetonide) patients? Please clarify the readers.

  1. The authors wrote:

“Eyes with cysts in the central 1000 microns or any intraretinal or subretinal fluid were also included in this study, independent of the CMT parameter.” Please clarify if the cysts were accompanied by hyperreflective dots. Please also confirm if those cysts were inflammatory or not, once patients with those inflammatory cysts may not respond to anti-VEGF treatments.

  1. Concerning tables and figures: The tables within the manuscript are adequate and provide a good understanding of the whole experience.

However, the readers would benefit from a resume table with the following information: genes, responder vs non-responder vs. sub responder, relation with CMT and BCVA, and significance.

Thank you.

Regards,

Author Response

Dear Reviewer,

Thank you for giving us the opportunity to revise our Manuscript ID: ijms-1633660: “Identifying genetic biomarkers predicting response to anti-vascular endothelial growth factor injections in diabetic macular edema”. We have incorporated the suggestions made by the reviewers. The revisions are highlighted in yellow in the revised manuscript. We have updated the manuscript as per the IJMS template. The revisions have been approved by all authors. Please see our detailed responses to the comments below. All line numbers refer to the revised manuscript file.

Reviewer 1 comments:

Comment 1: In the beginning please refer: to BCVA in ETDRS letters.

Response: As per the reviewer’s suggestion, we have made necessary changes in the manuscript. Updated: Line 22

Comment 2: “adjusting for the first three principal components, age, baseline CMT/BCVA, duration of diabetic retinopathy, and HbA1c – clarify adjustments performed.

Response: We accounted for the confounding effects these factors can have on the outcome measure by including them as covariates in the regression analysis. This has been described in detail in the methods section (Lines 345-347, 351-356) as has not been altered in the abstract section due to word count restrictions.

Comment 3: I suggest replacing the term collection for other more appropriate for the definition provided

Response: as per the reviewer’s suggestion, the word “collection” has been replaced by “accumulation”. Updated: Line 46

Comment 4: The abbreviation should be written in full the first time it appears: GC

Response: Full form of QC (quality control) updated: Line 69

Comment 5: Explain why these patients were excluded based on QC. Please clarify.

Response: The details of the reasons for and numbers of participants excluded at each QC step are provided in Supplementary Table 1. The QC steps are described in the materials and methods section (Lines 319-333, 342-343). A statement indicating where to find these details has been added to the Results section. Lines 70-72.

Comment 6: In Table 1 – Are the following values: Baseline BCVA (ETDRS letters), Final BCVA (ETDRS letters), Change Baseline CMT (microns), and Final CMT (microns) means? Please clarify.

Response: Values in Table 1 are presented as mean (standard deviation) for continuous variables or number (percentage) for categorical variables, as noted in the table footnote.

Comment 7: Are the mean variation BCVA and CMT statistically significant? Please add p values to table 1.

Response: Yes, the mean changes in BCVA and CMT are statistically significant. This has been clarified in the text (Lines 75-77) and in the Table 1 Footnote. We have also updated the details of statistical analysis for demographic data in the Methods section (Lines 370-374)

Comment 8: Correct to Figure 1A?

Response: Thank you. Correction made. (Line 102)

Comment 9: Even though this study is in the real-world environment, may you please clarify the protocol used at the hospital for the treatment of DME (including treatment regimen)?

Response: This was a multi-center clinic-based observational study. Treatment decisions for each patient were made by the treating physicians and vary between physician and clinic/hospital. There was no standardized approach to this. This is noted in the Materials and Methods: Phenotype section (Lines 285-288)

Comment 10: The authors define the following (line 336 to 340):

 “A functional responder was defined as 1) ≥5 ETDRS letters improvement from the baseline or 2) ≥15 ETDRS letters improvement from baseline BCVA. A functional non-responder was defined as ≥5 ETDRS letters loss from baseline. An anatomical responder was defined ≥10% reduction in CMT from the baseline.” Please clarify, if a patient does not lose or gain vision letters would this be a responder or non-responder? This definition should be clarified as does not follow any standard guidelines. Also please clarify the sub responder.

Response: We adopted these definitions of responders and non-responders based on definitions in relevant randomized clinical trials and anti-VEGF treatment guidelines. The references have been added to the manuscript (Lines 297-301). Each of these categories is compared to the remainder of the cohort, not to each other. As an example, participants with less than 5 letters improvement are included in the ‘non-responder’ group when being compared to responders with ≥5 letters improvement. Additional information has been added to the methods to better explain this (Lines 345-348).

For this study. we classified patients with no gain or loss in vision as non-responders as our definition for responder is an increase in either 5 letters or 15 letters.  In an ideal situation with a larger cohort, it would definitely be useful to explore this sub-group of participants as a separate cohort.

Comment 11: The authors wrote: “Exclusion criteria were: (a) any vitreoretinal surgery, systemic anti-VEGF therapy, or intra-ocular steroid in the six months preceding the initiation of anti-VEGF injection, (b) insufficient visibility of the fundus for retinal diagnosis, (c) incomplete follow-up data, and d) inability to give consent.” Did the authors exclude Iluvien implant (fluocinolone acetonide) patients? Please clarify the readers.

Response: As noted in the exclusion criteria (Line 278), patients receiving intraocular steroids (which includes Iluvien) were excluded.

Comment 12: The authors wrote: “Eyes with cysts in the central 1000 microns or any intraretinal or subretinal fluid were also included in this study, independent of the CMT parameter.” Please clarify if the cysts were accompanied by hyperreflective dots. Please also confirm if those cysts were inflammatory or not, once patients with those inflammatory cysts may not respond to anti-VEGF treatments.

Response: As this was a retrospective data collection, the full details of the OCT, such as mentioned by the reviewer, were unfortunately not available. This has been mentioned as a limitation of the study (Lines 249-250).

Comment 13: Concerning tables and figures: The tables within the manuscript are adequate and provide a good understanding of the whole experience. However, the readers would benefit from a resume table with the following information: genes, responder vs non-responder vs. sub responder, relation with CMT and BCVA, and significance.

Response: Thank you for the suggestion. A table as described would help with clarity, however, the significant findings we have generated in this preliminary study relate only to the quantitative change in vision and macular thickness and their association with SNPs (not specific genes). The information regarding the relationship between each significant SNP and these traits are given in Table 2 where the Beta values indicate the magnitude and direction of the effects. We do not find any significant results when considering the division of participants into responders or non-responders with these or other SNPs and therefore cannot provide the information requested.

Thank you.

Kind Regards

Reviewer 2 Report

GWAS analysis was performed on DNA samples of patients with DME treated with anti-VEGF therapy. In total 220 patients were included. SNP genotypes were correlated with changes in BCVA and CMT during a period of 12 months. Although the study group is relatively small for GWAS study and a reference cohort is necessary to validate these findings, this is the first study investigating genetic variance in relation to anti-VEGF therapy in DME. Therefore, this study is interesting and relevant for the field as a first step in understanding individually based differences in therapy response in DME patients.

Extra attention is needed for the following:

Due to the low number of patients, care must be taken in the interpretation of the SNPs found. The authors are fully aware of this, as evidenced by their comments on the subject in the discussion section. However, a further distinction of probable outliers in Figure 1 could be made. A characteristic of truly correlated SNPs is that they do not hang out alone “in the sky”, but are usually followed by a series of SNPs with a smaller p-value. Therefore, regarding the CMT, the SNP on chromosome 6 is likely to be a coincidental hit and an outlier, whereas the SNP on chromosome 12 is likely to be reliable. This distinction could be more clearly stated in the text.

Author Response

Dear Reviewer,

Thank you for giving us the opportunity to revise our Manuscript ID: ijms-1633660: “Identifying genetic biomarkers predicting response to anti-vascular endothelial growth factor injections in diabetic macular edema”. We have incorporated the suggestions made by the reviewers. The revisions are highlighted in yellow in the revised manuscript. We have updated the manuscript as per the IJMS template. The revisions have been approved by all authors. Please see our detailed responses to the comments below. All line numbers refer to the revised manuscript file.

Reviewer 2 comments

Extra attention is needed for the following:

Due to the low number of patients, care must be taken in the interpretation of the SNPs found. The authors are fully aware of this, as evidenced by their comments on the subject in the discussion section. However, a further distinction of probable outliers in Figure 1 could be made. A characteristic of truly correlated SNPs is that they do not hang out alone “in the sky”, but are usually followed by a series of SNPs with a smaller p-value. Therefore, regarding the CMT, the SNP on chromosome 6 is likely to be a coincidental hit and an outlier, whereas the SNP on chromosome 12 is likely to be reliable. This distinction could be more clearly stated in the text.

Response: The reviewer makes a pertinent point and as such, we have made relevant changes to the discussion as suggested (Lines 185-189).

Thank you.

Kind Regards

Round 2

Reviewer 1 Report

Dear authors,

Thank you for addressing all queries raised.

I have no further comments.

Regards,